# Comparison of Methods for Quantifying Extracellular Vesicles of Gram-Negative Bacteria

**DOI:** 10.3390/ijms242015096

**Published:** 2023-10-11

**Authors:** Chanel A. Mosby, Natalia Perez Devia, Melissa K. Jones

**Affiliations:** Microbiology and Cell Science Department, Institute of Food and Agricultural Sciences, University of Florida, Gainesville, FL 32611, USA; c.mosby.haundrup@ufl.edu (C.A.M.); natalia.perez02@outlook.com (N.P.D.)

**Keywords:** bacterial extracellular vesicle, outer membrane vesicles, nanoparticle tracking analysis, vesicle quantification, Qubit, NanoOrange, microBCA, FM 4-64, NTA

## Abstract

There are a variety of methods employed by laboratories for quantifying extracellular vesicles isolated from bacteria. As a result, the ability to compare results across published studies can lead to questions regarding the suitability of methods and buffers for accurately quantifying these vesicles. Within the literature, there are several common methods for vesicle quantification. These include lipid quantification using the lipophilic dye FM 4-64, protein quantification using microBCA, Qubit, and NanoOrange assays, or direct vesicle enumeration using nanoparticle tracking analysis. In addition, various diluents and lysis buffers are also used to resuspend and treat vesicles. In this study, we directly compared the quantification of a bacterial outer membrane vesicle using several commonly used methods. We also tested the impact of different buffers, buffer age, lysis method, and vesicle diluent on vesicle quantification. The results showed that buffer age had no significant effect on vesicle quantification, but the lysis method impacted the reliability of measurements using Qubit and NanoOrange. The microBCA assay displayed the least variability in protein concentration values and was the most consistent, regardless of the buffer or diluent used. MicroBCA also demonstrated the strongest correlation to the NTA-determined particle number across a range of vesicle concentrations. Overall, these results indicate that with appropriate diluent and buffer choice, microBCA vs. NTA standard curves could be generated and the microBCA assay used to estimate the particle number when NTA instrumentation is not readily available.

## 1. Introduction

Membrane-enclosed extracellular vesicles (EVs) are ubiquitously produced by bacteria. Those produced by Gram-negative bacteria are generally referred to as outer membrane vesicles (OMVs) or inner-outer membrane vesicles, depending on the mechanism of vesicle biogenesis that resulted in vesicle formation [1]. The content of bacterial extracellular vesicles (bEVs) can vary based on the mechanism of biogenesis, but all bEVs have been shown to contain both lipids and proteins derived from their parental bacterium [2,3,4]. As a result of the predictability of protein and lipid cargo, measuring these components is frequently used as a surrogate for quantifying the number of vesicles isolated from bacterial cultures. In addition to these indirect quantification methods, direct vesicle measurement is also commonly employed to quantify bEVs using techniques such as nanoparticle tracking analysis (NTA) [5,6,7]. For all vesicle measurement methods, the accuracy and reproducibility of vesicle quantification can vary based on the method used and how vesicles were prepared prior to quantification.

A variety of different methods are used to isolate extracellular vesicles, which vary based on vesicle type (i.e., exosome, nanovesicle, OMV, etc.) and the downstream applications the vesicles for which the vesicles are intended [8,9]. One of the most widely used techniques to isolate vesicles is ultracentrifugation-based methods. These methods generally consist of serial centrifugation steps that increase in both speed and time, allowing for the concentration and isolation of EVs from the sample matrix. Another commonly used method of EV isolation is size exclusion chromatography, which isolates EVs from a sample based on size exclusion limits. Precipitation methods also exist, which use organic solvents to remove soluble proteins, leaving behind the concentrated vesicles. However, this step alone can lead to the co-precipitation of other materials, so precipitation methods are often combined with other isolation steps to reduce that possibility [9]. If the desired EVs have known markers, immunoaffinity methods can be used to separate out the EVs of interest, but this method is limited by the need to have specific biomarkers that are membrane-bound and specific antibodies that target the biomarker [9,10]. Finally, microfluidic systems have also been applied to EV isolation and allow for both size and immunoaffinity-based selections; however, these systems come with increased upfront cost compared to the other methods [9,10]. These methods vary in the amount of yield and the specificity of the isolated EVs. In general, precipitation methods are considered to give the highest yield but the lowest specificity [10]. Ultracentrifugation and size exclusion chromatography methods give a medium yield and medium specificity, while immunoaffinity and microfluidic systems result in lower yields but the highest specificity of the various isolation methods [10].

After isolation, there are a variety of methods used to quantify EVs. NTA is increasingly used to quantify vesicles and calculates the total number of particles in a sample based on the Brownian motion of the particles and the principles of light scattering [11]. This method is commonly used for determining bEV particle numbers as well as vesicle diameter. For many NTA instruments, there is a level of operator expertise that is required for optimal results, which depend on the user having the right combination of camera level and focus, sample dilution, and particle detection threshold [11]. Variation in these parameters can lead to variability in determining vesicle concentration, and there are limited methods available to validate the NTA results. In addition, NTA equipment is costly, putting it out of reach for some laboratories and providing a financial challenge to the solution of having multiple technicians run the same sample to reduce variability due to operator bias. As a result, more affordable methods that measure proteins or lipids are widely used as an indirect means to quantify bacterial vesicles.

A commonly used and widely available method for bEV measurement is the microBCA (bicinchoninic acid) assay for protein quantification. This colorimetric-based assay uses Cu^2+^ ions, which interact with proteins in a sample, inducing a color change, which is directly proportional to the concentration of protein in a sample [12]. The microBCA assay is sensitive and suitable for measuring low protein concentrations, ranging from 0.5 to 20 µg/mL. Another option for quantifying protein amounts in bEVs is the Qubit protein assay [13,14]. This fluorometric assay uses dyes to bind to protein where the dye subsequently fluoresces. Since target dyes are unable to bind to contaminants, this assay minimizes the possibility of overestimation of the protein concentration. However, this assay has a higher limit of detection than microBCA (12.5 µg/mL). NanoOrange is another fluorometric assay for quantifying the protein concentration of vesicles. The NanoOrange dye binds to amino acids. This assay is the most sensitive of the three discussed, able to detect protein concentrations as low as 10 ng/mL [15]. In addition to protein assays, lipid detection assays are also sometimes employed to quantify extracellular vesicles [16]. One such assay uses the FM 4-64 lipophilic styryl compound that is often used in studies of plasma membranes and vesiculation due to its ability to intensely fluoresce when inserted into the outer membrane of a phospholipid bilayer.

Given the diversity of methods used for the quantification of bEVs, the primary aim of this work was to directly compare the reliability of buffers and diluents on protein concentration measurements and particle quantification of bEVs. Using OMVs derived from the commensal bacterium *Enterobacter cloacae*, the results showed that microBCA produced the most reliable protein quantification while the results using the Qubit assay varied greatly with the lysis buffer and vesicle diluents used. The results also showed that microBCA protein measurements were the most strongly correlated with NTA particle number, particularly when OMVs were resuspended in TE (Tris-EDTA) buffer, making it the best substitute for OMV quantification if NTA is not available.

## 2. Results

### 2.1. Impact of Lysis Buffers on Protein and Lipid Quantification

The measurement of protein content is commonly used for quantifying bEVs. However, not only are different methods employed by a variety of laboratories, but also different types of lysis buffers. RIPA (radioimmunoprecipitation assay) buffer and Triton-X are two of the most commonly used buffers for measuring the vesicle protein concentration. These are often made in large quantities and used over time, but buffer preparation and freshness are rarely reported. Since lysis efficiency could impact protein measurement, we tested whether buffer type and buffer age affected OMV protein concentration measurements using three of the most common assays: microBCA, Qubit, and NanoOrange. OMVs were isolated from pure cultures of the commensal bacterium *E. cloacae* and resuspended in 1× protease inhibitor (PI) diluted in dPBS (Dulbecco’s phosphate-buffered saline), divided into equal portions, and then treated with fresh vs. old (4.5 months) buffers prior to protein quantification. A portion of the OMVs was also left untreated (no lysis buffer) prior to protein measurement as a control. The results showed that, for all methods of protein measurement, the type and age of the lysis buffer did not significantly alter protein quantification (Figure 1A). Interestingly, the microBCA readings for the “no lysis buffer” controls were also not significantly different from the concentration readings of the lysed samples. This pattern also largely held when NanoOrange was used to measure protein concentration. For NanoOrange readings, the protein quantities trended lower in the absence of lysis buffer, but the variability in the readings using this assay precluded statistically significant differences being observed with all buffers except the fresh Triton-X (*p* < 0.01; Figure 1A). Conversely, when the Qubit assay was used to measure protein concentration, the use of lysis buffer did contribute to significantly increased protein measurements (*p* < 0.0001) compared to no lysis controls for all buffers tested (Figure 1A).

The quantification of vesicles by measuring lipid concentration is also frequently found in the published literature. Therefore, using the same bEV samples mentioned above, we quantified their lipid content using the FM 4-64 assay. Similar to what was observed with protein measurement, neither lysis buffer type nor lysis buffer age impacted the lipid concentration readings (Figure 1B). However, the use of lysis buffer with this assay was critical for obtaining measurable lipid concentrations where the lysis of vesicles allowed for significantly (*p* < 0.01) higher lipid readings compared to readings without lysis buffer (Figure 1B).

Ultimately, these results showed that, regardless of the assay selected, the type and age of lysis buffer are not critical variables when measuring OMV protein and lipid concentrations and will not significantly affect the protein or lipid readings. Lysing of vesicles is standard practice but was not absolutely required for obtaining protein concentrations using microBCA. However, the lysis of OMVs did always yield higher protein concentrations with this assay (although not significantly higher), indicating that vesicle lysis is still needed to accurately measure total protein concentrations in OMVs. Finally, of the three protein assays that were compared, microBCA yielded consistently lower protein concentrations while NanoOrange displayed the greatest variability between samples. Another observation made when comparing the protein measurement assays was the large differences in protein concentration obtained with each assay. For microBCA, the average protein concentration was 81.6 µg/mL, for NanoOrange it was 706.8 µg/mL, and for Qubit it was 3128.4 µg/mL. Since Qubit and microBCA are more commonly used than NanoOrange, and since Qubit and NanoOrange yielded far higher protein concentrations compared to microBCA, we used microBCA and Qubit assays to further explore the reasons behind these protein concentration discrepancies.

### 2.2. Impact of Diluent on Vesicle Quantification

In light of the discrepancies in protein concentration observed between microBCA and Qubit assays, we next investigated whether the diluent used to resuspend OMVs would alter the protein measurement. As with lysis buffer, the diluents used to resuspend isolated bEVs also vary within the literature. Three of the primary diluents used are TE, dPBS, and dPBS + PI [17,18,19]. PI is often included in diluents with the purpose of increasing vesicle stability and preserving the vesicle protein content. Vesicles were extracted from *E. cloacae* and diluted in dPBS only, dPBS + PI, or in TE. Since lysis buffers were previously compared, this study boiled OMVs as a means of vesicle lysis prior to protein quantification so only the chemicals of the diluents would be present during measurement. For each sample, protein amounts were quantified using microBCA and Qubit, and the particles were enumerated using NTA. The results showed that the diluent had no impact on OMV quantification for microBCA and NTA, but the presence of PI resulted in a dramatic increase in protein measurement when Qubit was used (Figure 2A). Often, buffers that result in large background signal can be used as blanks to account for the increased signal. However, for Qubit measurements, when blanking with dPBS + PI, quantification for the blank was repeatedly the same as or larger than the readings obtained when measuring the sample, indicating that PI overwhelms the instrument, generating results that are not interpretable. This can be seen when the diluents alone were measured (Figure 2B, right panel) where Qubit measurements approximated what is observed for OMVs resuspended in dPBS + PI (Figure 2A). These results indicate that Qubit is not usable for the protein quantification of OMVs when the diluent used for vesicle resuspension contains PI. Diluents were also measured using microBCA, and in this assay, PI was also detectable (Figure 2B, left panel), but the detectable range of this assay is such that the signal generated by the buffer can be accounted for as a blank and still yield reliable protein concentrations.

### 2.3. Determining Correlation between Protein and Particle Measurement Assays

In addition to comparing the individual vesicle quantification methods, we also compared the ability of these assays to quantify vesicles over a concentration range. Moreover, since NTA instrumentation is not available to all vesicle labs, we set out to determine if there was a correlation between protein concentration and particle numbers between these samples. To simulate higher and lower OMV concentrations, samples were serially diluted, and these dilutions were compared among the assays. These experiments were performed using the three diluents discussed above and employed microBCA and Qubit assays for protein measurement. When comparing protein measurements using Qubit to NTA readings, dilution of the samples reduces OMV protein concentrations at or below the limit of detection for both dPBS and TE diluted vesicles, therefore, we did not include OMVs resuspended in dPBS + PI since we previously established that this diluent results in inaccurate protein readings with Qubit (Figure 2). Comparing the correlation of 1× OMV samples to NTA, the results showed that resuspension in TE displayed a much stronger correlation with NTA than resuspension in dPBS (Figure 3).

When comparing microBCA readings to NTA measurements, an overall strong positive correlation (R^2^ = 0.88, 0.93, and 0.86) was found between NTA and microBCA readings when dPBS, dPBS + PI, and TE were used as diluents, respectively (Figure 4A,C,E). When looking at the correlation between microBCA and NTA at different OMV concentrations, strong correlations were also seen with all buffers when vesicles were undiluted (1×; Figure 4B,D,F). However, the dilution of OMVs was found to generally decrease the correlation between the assays in a diluent-dependent fashion. For OMVs resuspended in dPBS (both with and without PI), as the vesicles were diluted, the correlation with NTA particle readings also decreased (Figure 4B,D). However, when OMVs were resuspended and diluted in TE, the correlation between the two assays remained high (Figure 4F). Therefore, as with Qubit, TE provides a stronger correlation between protein concentration and particle count when microBCA is used. These results indicate that at higher OMV concentrations, any of these diluents will result in protein measurements that correlate well with particle number. However, if vesicle concentrations are lower, the use of TE will provide stronger correlations between protein concentration and particle count. Therefore, when quantifying OMV extracts suspected to be at low vesicle concentrations, resuspension in TE is most likely to allow microBCA measurements to provide the nearest correlation to NTA particle counts.

## 3. Discussion

The publication of research examining the biological mechanisms of and practical applications for bacterial extracellular vesicles is increasing. Vesicles from a wide variety of bacteria are being explored for their roles in bacterial and viral infections [20,21,22,23] and their use as a delivery mechanism for disease treatment or prevention [24,25]. Considerable amounts of research have explored the role of bacterial vesicles in the survival and pathogenicity of the parental bacterium [26,27,28,29]. More recently, it has been shown that vesicles produced by commensal bacteria can influence viral infection, although the impact on infection appears to be virus and potentially location-dependent within the host [20,23,25]. Within the host, bacterial vesicles are capable of migrating throughout the body and can also stimulate host immune responses [30,31]. For these reasons, they are also being explored as novel vaccine platforms. Vesicle vaccines used to elicit a response against the parental bacterium are being explored for pathogens such as Neisseria meningitidis [32]. Bacterial vesicles are also being engineered to target other bacterial and viral pathogens, such as Chlamydia trachomatis and SARS-CoV-2 [24,33].

However, the methods for quantifying EVs vary widely within the literature. The present study was performed to assess differences between the critical steps of OMV measurement and compared how different means of lysis and vesicle resuspension altered the output of these quantification assays. When evaluating protein concentration using Qubit, microBCA, and NanoOrange assays, the results demonstrated that the impact of buffer and diluents was assay-specific. When the Qubit assay was used, buffers and diluents resulted in strong interference with protein measurement. In particular, the inclusion of a protease inhibitor strongly masked the protein signal of OMVs in this assay. The major advantages of the Qubit method are the speed of the assay, low variability across measurements, and its relatively low cost of use, making it ideal for quantifying OMVs prior to downstream applications. However, users should avoid using traditional lysis buffers or adding protease inhibitors to their samples with the Qubit assay as these result in artificially high readings. It should also be noted that when dPBS alone or TE was used as diluents, each of these yielded similar protein outputs, and the Qubit assay was able to detect lower OMV concentrations, making it useful for quantifying protein amounts in diluted OMV samples. For the microBCA assay, the inclusion of PI in the dPBS diluent increased the protein concentrations, as seen with Qubit, but the detectable range of microBCA allows for the subtraction of diluent blank concentrations. PI is most often seen with downstream proteomics studies where preventing degradation by proteases is a concern. Given that PI contributes to much higher readings compared to the other diluents, including proper blanks and controls is crucial for protein quantification accuracy when using this diluent.

NanoOrange was also tested for the protein quantification of OMVs. This assay is less commonly used for OMV quantification but is relatively cost effective. Like Qubit and microBCA, NanoOrange readings were not impacted by the type of lysis buffer used. However, among the protein assays, NanoOrange displayed the greatest variability between samples. Lipid measurement using FM 4-64 was also explored as an alternative to protein quantification. Lipid measurements were also consistent despite the age and type of buffer used. While the measurement is easy to perform and the results were consistent, its use in the published literature often varies in terms of concentration, which, along with a lack of methodology details, makes it more difficult to consistently compare results across papers. It is also used less often than microBCA or particle counts for bacterial extracellular vesicle concentration analysis.

Not surprisingly, when NTA was used to quantify OMVs, there was little to no difference between the particle numbers, regardless of the diluent (i.e., dPBS, dPBS + PI, TE; Figure 4). However, unlike the protein measurements, NTA did not accurately account for the dilution of samples. This result was surprising as NTA has been shown to detect dilutions of EVs from other species [34]. This discrepancy with our data may be a limitation of the method or due to the non-homogeneous distribution of OMVs within a sample. While previous characterizations of *E. cloacae* bEVs have shown that these vesicles are fairly homogeneous [2,7], bEVs can have a wide range of identities and sizes depending on the bacterium they are isolated from, the bacterial growth phase from which they are harvested, and the mechanism of biogenesis through which they are formed [35,36,37]. As a result, these differences may lead to inconsistency with previously published data, demonstrating the reliability of NTA. It is also worth noting that even in publications where NTA quantification tracks with dilution, the accuracy of the measurements is dependent on vesicle concentration. Therefore, while NTA has been widely demonstrated and confirmed as a reference method for vesicle enumeration, its suitability may need to be empirically determined for different types of bEVs. An alternative to NTA for quantifying vesicles is a newly developed technique called Interferometric Light Microscopy (ILM). This method uses Brownian motion to detect particles similar to NTA. However, it uses an LED, rather than lasers, to illuminate the particles and a transmission bright-field microscope to detect the particles, which lowers the cost of this technique compared to NTA. Several studies have demonstrated the ability of this method to detect large viruses and vesicles [38,39,40]. However, the accuracy of vesicle measurements using this method decreases with decreasing particle size [41]. As mentioned above, NTA measurement requires the use of costly instrumentation that is subject to user variability. Currently, the Nanosight instrument used in this study costs over USD 94,000. As a result, the measurement of particle numbers with this method may not be financially feasible for all research laboratories. For this reason, we tested the correlation between protein concentration and particle quantification using microCBA and Qubit assays (Figure 3 and Figure 4). Qubit fluorimeters are priced under USD 4000, and protein measurement reagents cost about USD 0.80 per sample. Protein measurements for microBCA are performed in test tubes or in a microplate format and read using standard spectrophotometers. The cost for reagents is approximately USD 0.53 per sample. The results showed that microBCA showed the highest correlation with NTA across all dilutions compared to Qubit and that correlations were strongest when OMVs were resuspended in TE. Ultimately, this demonstrates that if NTA instrumentation is regularly unavailable, microBCA vs. NTA standard curves could be developed and employed when using microBCA to estimate particle number.

## 4. Materials and Methods

### 4.1. Bacterial Strain and Growth Conditions

The bacterium *Enterobacter cloacae* (ATCC 13047) was used for all experiments. *E. cloacae* was cultivated in Luria Bertani (LB) medium with 1% NaCl (sodium chloride) under aerobic conditions at 37 °C with constant shaking (220 rpm).

### 4.2. Generation and Isolation of Bacterial Membrane Vesicles

Bacterial extracellular vesicles were generated as previously described with some modifications [7]. Briefly, bacteria were grown up to stationary phase in 120 mL of LB broth prior to being washed twice with 1× PBS and concentrated into 5 mL of PBS. The OD_600_ reading was taken to adjust the bacterial concentration to 10^8^ cell/mL based on previous growth curves. Plating with serial dilution was used to confirm the bacterial count. The bacteria were then inoculated in two 60 mL flasks of fresh LB and grown for 12 h at 37 °C. The contents of each flask were centrifuged at 2000× *g* for 20 min at 4 °C to pellet out bacterial cells, after which the supernatant was ultracentrifuged at 25,000× *g* for 20 min at 4 °C. The resulting supernatants were filter sterilized with a 0.22 μm filter into new ultracentrifuge tubes and ultracentrifuged at 150,000× *g* for 2 h at 4 °C. The vesicle pellets resulting from the two flasks were resuspended in dPBS (Cytivia, Marlborough, MA, USA) and combined prior to a final ultracentrifugation spin at 150,000× *g* for 2 h at 4 °C before a final resuspension in 500 μL of diluent. Vesicles were stored at 4 °C before being used for quantification assays.

### 4.3. Lysis Buffers and Diluents

To lyse the cells, various lysis buffers were tested with each method. The buffers prepared included RIPA and Triton X (Sigma Aldrich, St. Louis, MO, USA). The buffers were used freshly made while others were used 4.5 months after preparation to determine if lysis buffer age affected lysing. Diluents used in this study included dPBS, dPBS with 1× protease inhibitor cocktail (Thermo Fisher no. A32955, Waltham, MA, USA), or TE buffer (10 mM Tris-HCl, 0.1 mM ethylenediaminetetraacetic acid (EDTA)).

### 4.4. Protein Quantification Assays

The protein quantifications were performed using the Micro BCA™ Protein Assay (Thermo Fisher Scientific), the Qubit™ Protein assay kit (Thermo Fisher Scientific), and the NanoOrange™ Protein Quantification Kit (Thermo Fisher Scientific). A standard curve was prepared for both the microBCA and NanoOrange assays using Bovine Serum Albumin (BSA), with final concentrations (in μg/mL) of 100, 50, 40, 30, 20, 10, 5, 2, and 0. The OMV sample was diluted 1:3 dilution before use in the microBCA and NanoOrange assays per manufacturer’s instructions. The Qubit assay kit provides 3 standards for calibration, which were used followed by the BSA standards above run as samples to confirm the concentration of the BSA standards. The OMV samples were not diluted before running the assay on the Qubit™ 4 Fluorometer per manufacturer’s instructions.

### 4.5. Lipid Quantification Assays

Lipid quantification was performed using FM 4-64 (Molecular Probes, Eugene, OR, USA), where OMV samples diluted 1:3 were incubated with 5 μg/mL in PBS for 10 min at 37 °C. Vesicles without FM 4-64 dye and FM 4-64 dye-only samples were also run as controls. Results were read on a spectrophotometer with an excitation wavelength of 506 nm and an emission of 750 nm.

### 4.6. Nanoparticle Tracking Analysis (NTA)

OMVs were diluted 1:100 in filtered autoclaved water and loaded onto NanoSight NS300 (Malvern Panalytical, Malvern, UK). For each of at least three biological samples, the particle size and quantity were recorded for 60 s per technical replicate, with five technical replicates.

### 4.7. Effect of OMV Dilution on Protein and NTA Quantification

To determine the correlation between OMV protein concentration and particle number, microBCA, Qubit, and Nanosight assays were performed after serial dilution of harvested OMVs. Due to the large quantity of OMVs needed to perform these assays in parallel, the OMV isolation was slightly modified in that the outgrowth stage included four 60 mL flasks instead of two. Following this step, the OMV isolation followed the protocol outlined above including combining the isolated vesicles prior to the final spin. In addition, lysis buffers were not used. Instead, OMVs were lysed via boiling at 100 °C for 10 min for protein quantification only. An aliquot of OMVs was removed for Nanosight analysis prior to boiling.

### 4.8. Data Analysis and Statistics

Resultant values from the respective assays were loaded into dataframes in RStudio (Version 2023.06.1 + 524 with R version 4.3.1), where the packages tidyverse v1.2.1 and rstatix v0.7.2 were used to run the statistical tests, including one-way and two-way ANOVAs, followed by Tukey’s multiple comparison test as well as *t*-tests. Graphs were made with the ggpubr v0.6.0 R package [42].

## Figures and Tables

**Figure 1 ijms-24-15096-f001:**
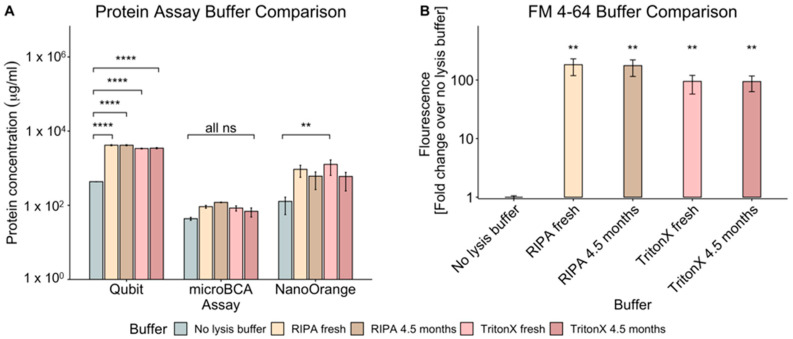
Impact of lysis buffer type and age on OMV protein and lipid quantification. OMVs isolated from *E. cloacae* were resuspended in dPBS with PI and then treated with lysis buffers of varying age and composition before analysis with protein assays (Qubit, microBCA, or NanoOrange) or with the lipophilic dye, FM 4-64. (**A**) Comparison of protein concentration readings of OMVs using Qubit, microBCA, or NanoOrange assays and with a no-lysis-buffer control, RIPA, or TritonX of 4.5 months of age or fresh. The statistical significance was found using Tukey’s pairwise comparison tests after a two-way ANOVA. (**B**) Comparison of fresh or 4.5-month-old RIPA or Triton-X buffers and a no-lysis-buffer control on the emitted fluorescence on OMVs incubated with FM 4-64 lipophilic dye. *t*-tests were used for pairwise comparisons to determine statistical significance. For ease of visualization, the asterisks show the result of different buffers against the no lysis buffer control for the FM 4-64 graph. Asterisks represent adjusted *p* values: **, *p*  ≤  0.01; ****, *p*  ≤  0.0001. ns = not significant Error bars display SEM, *n*  =  4.

**Figure 2 ijms-24-15096-f002:**
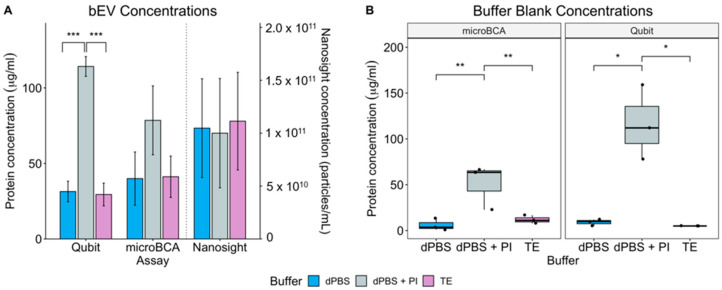
Impact of diluent on OMV quantification. (**A**) OMVs isolated from *E. cloacae* were resuspended in different diluents (dPBS, dPBS + protease inhibitor (PI), or TE buffer) prior to protein quantification via the Qubit or microBCA assays as well as particle count on the Nanosight. Protein quantification values in μg/mL for the Qubit and microBCA assays can be read on the left *y*-axis, while the particle counts in particles/mL can be found on the right *y*-axis for Nanosight readings. (**B**) The same diluents were run alone without any OMVs for the two protein assays (Qubit and microBCA). Statistical significance was found using Tukey’s pairwise comparison tests after a two-way ANOVA. Asterisks represent adjusted *p* values: *, *p*  ≤  0.05; **, *p*  ≤  0.01; ***, *p*  ≤  0.001. Error bars display SEM, *n*  =  3.

**Figure 3 ijms-24-15096-f003:**
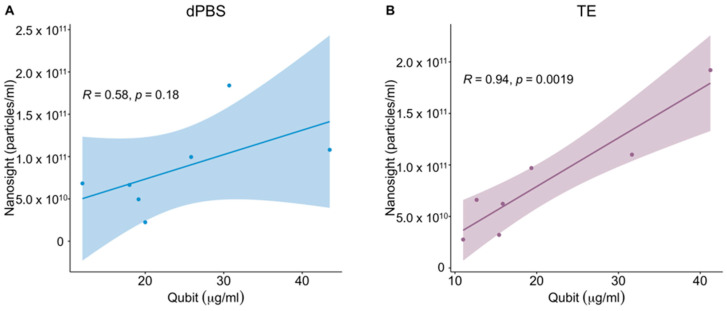
Correlation between NTA and Qubit measurement of OMVs in differing buffers. OMVs isolated from *E. cloacae* were resuspended in either (**A**) dPBS or (**B**) TE. The OMVs were then analyzed using Qubit for protein quantification and with NTA for particle count. A linear regression line and 95% confidence interval are displayed on the plots as well as the Pearson’s correlation coefficient. *n* = 3 biological replicates (2 technical replicates for each experiment).

**Figure 4 ijms-24-15096-f004:**
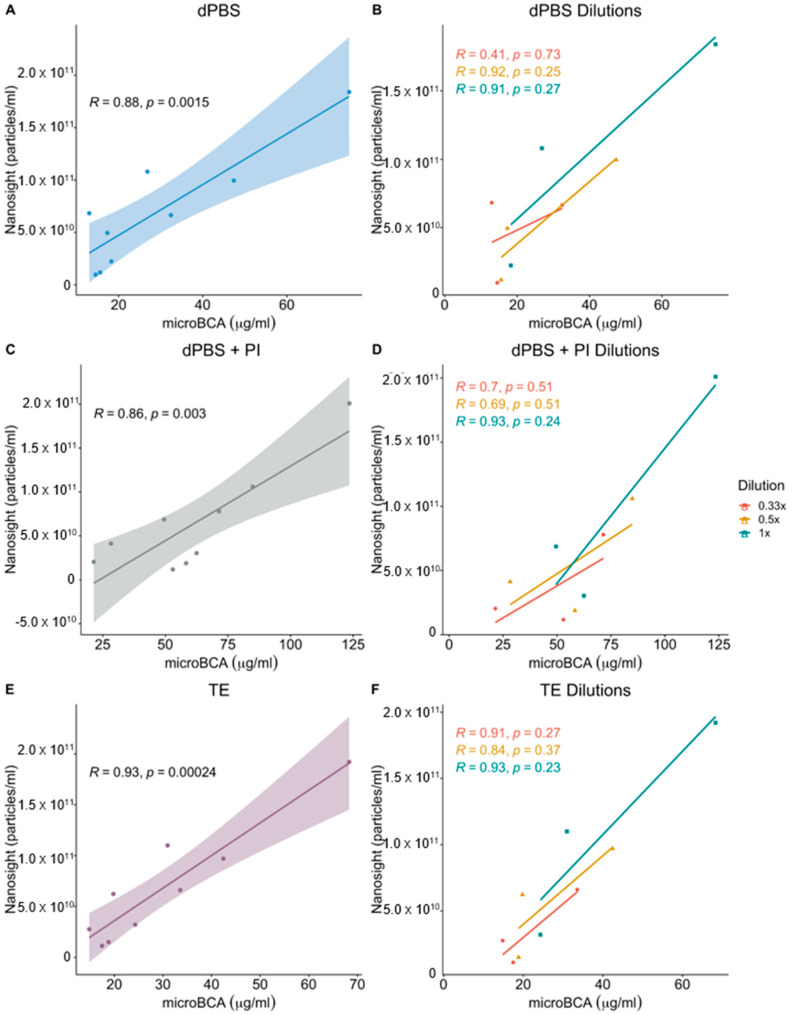
Correlation between NTA and microBCA measurement of OMV samples in differing buffers. OMVs isolated from *E. cloacae* were resuspended in (**A**) dPBS, (**C**) dPBS + PI, or (**E**) TE before serial dilution for 1:1, 1:2, and 1:3 dilutions with the respective diluents, (**B**,**D**,**F**). The OMVs were then analyzed using microBCA for protein quantification and with NTA for particle count. A linear regression line and 95% confidence interval are displayed on the plots as well as the Pearson’s correlation coefficient. *n* = 3 biological replicates (3 technical replicates for each experiment).

## Data Availability

Data will be made available in the publicly available repository Open Science Framework at osf.io.

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
