# Peer review of "Comparison of Methods for Quantifying Extracellular Vesicles of Gram-Negative Bacteria"

_ijms, 2023, doi:10.3390/ijms242015096_

Round 1
Reviewer 1 Report
The article from Mosby et al proposes an analysis of the methods of quantification of EVs produced by Gram-negative bacteria.
I think this work is very innovative and adapted for publication in International J of Molecular Sciences.
My major question is about the size of EVs. It is now known that the diameter of EVs produced by bacteria can be sometimes homogeneous, and sometimes very heterogeneous, according to the species and even sometimes according to a specific strain. As a result, the correlation presented in this paper between the numeration of EVs (here determined by NTA) and the protein quantification in an EV sample, can be severely different from one species/strain to another. I think the authors should at least recall in the discussion that this work was performed on EVs from Enterobacter cloacae, and that the correlations presented in figures 3 and 4 cannot be easily extrapolated to EVs produced by other species/strains. This point does not impair the quality of the work and the general conclusions, however.
I also wondered why the authors did not include the Bradford assay in this work? It is another very common method to quantify proteins.
Figure 3, why the authors did not studied the correlation between NTA and Qubit with different dilutions, as performed in Figure 4 with microBCA? I really think this should be done for paper coherence.
L235-236, I do not understand why the authors claim that with Qubit assay, “buffers and diluents resulted in strong interference with protein measurements”? According to Figure 1A, the Qubit assay gave very similar results irrespective of the buffer used.
L261-262, I cannot understand where are the data studying NTA results according to different diluents. Please explain and add the corresponding data.
As a minor point, I recommend the authors to explicit any acronyms, including RIPA, TE, BCA, dPBS…
Author Response
Dear Reviewer,
We would like to thank you for your constructive comments and suggestions which have served to improve the content, quality, and clarity of the manuscript. The revised manuscript contains the text revisions and additions as was requested. Specifically, we improved the quality of the introductory material, explanations about the various assays and their costs, as well as added clarifying information to the results and discussion sections. We appreciate the insight and perspective you provided and have addressed each of your specific concerns below.
My major question is about the size of EVs. It is now known that the diameter of EVs produced by bacteria can be sometimes homogeneous, and sometimes very heterogeneous, according to the species and even sometimes according to a specific strain. As a result, the correlation presented in this paper between the numeration of EVs (here determined by NTA) and the protein quantification in an EV sample, can be severely different from one species/strain to another. I think the authors should at least recall in the discussion that this work was performed on EVs from Enterobacter cloacae, and that the correlations presented in figures 3 and 4 cannot be easily extrapolated to EVs produced by other species/strains. This point does not impair the quality of the work and the general conclusions, however.
We strongly agree with the reviewers point that this data, while informative, can’t necessarily be directly extrapolated to EVs from other bacteria. We have added language to the text that discusses what we know about E. cloacae EVs and also noted within the manuscript that correlations for other types of bEVs will need to be empirically determined (Lines 310-322).
I also wondered why the authors did not include the Bradford assay in this work? It is another very common method to quantify proteins.
The study was initiated to compare methods used by the Jones lab for vesicle quantification to try and find a correlating surrogate for NTA. Our nanosight was down for 3 months which was the precipitating event that caused us to look for other measures that might serve as a good surrogate if that were to happen again. This study was initiated by an undergraduate researcher (Natalia Perez) and overseen/finalized by a graduate student (Chanel Mosby). Both students have graduated and left the lab which is why it wasn’t expanded beyond the assays that we typically or previously used.
Figure 3, why the authors did not studied the correlation between NTA and Qubit with different dilutions, as performed in Figure 4 with microBCA? I really think this should be done for paper coherence.
These experiments were performed, and we found that the lower limit if detection for Qubit is ~12.5ug/mL. Therefore, when bEV samples were diluted, protein concentrations would decrease to be either at or below that limit leading to high variability and data that was not interpretable. This information can be found in lines 215-223.
L235-236, I do not understand why the authors claim that with Qubit assay, “buffers and diluents resulted in strong interference with protein measurements”? According to Figure 1A, the Qubit assay gave very similar results irrespective of the buffer used.
Figure 1A was performed with OMVs where the final resuspension step included protease inhibitor, in this figure the average protein concentrations for microBCA, NanoOrange, and Qubit were 81.6 µg/mL, 706.8 µg/mL, respectively. As seen in Figure 2, the addition of protease inhibitor had a marked effect on the Qubit readings, but the lysis buffers used in Figure 1A further enhanced that protein measurement reading. The presence of protease inhibitor is mentioned in the results text (lines 112-113) and in the legend of Figure 1. The issues with protease inhibitor and Qubit was also added to the discussion (lines 278-279)
L261-262, I cannot understand where are the data studying NTA results according to different diluents. Please explain and add the corresponding data.
The data examining NTA with different diluents and dilutions can be found in Figure 4. We have added this clarifying point to the discussion section (lines 307-320).
As a minor point, I recommend the authors to explicit any acronyms, including RIPA, TE, BCA, dPBS…
We have gone through the manuscript and addressed any acronyms that were not previously defined.
Reviewer 2 Report
The authors compared the assessment of protein and lipid content of isolates from conditioned media of bacteria with nanotracking device – derived measurements of the number density of small cellular particles in different situations. They claim that regardless of the assay selected, the type and age of lysis buffer and of diluent are not critical variables when measuring protein and lipid concentrations in samples. However, Qubit is not usable for protein quantification of EVs in the presence of PI. They suggest that assessment of protein content could be used to estimate the amount of EVs in the samples. Please consider the comments below.
As there are many types of small cellular particles in the biological samples (e.g. membrane-enclosed vesicles, antibody complexes, lipoproteins, extracellular nucleic acids, cell-engineered particles such as flagella hairs and scales, etc), the authors should state at the beginning what they mean by “extracellular vesicles”. Lipids and proteins may be involved in different types.
Did the authors visualize the samples to make sure which types of small cellular particles are present?
Line 261. NTA measurements did not reproduce well the proportion of EVs related to the dilution of samples. It should be mentioned that EVs do not have fixed identity and can be transformed in number, size and shape if diluted. Please comment how safe is to consider NTA as the reference method.
Line 267. The authors state:” For this reason, we tested the correlation between protein concentration and particle quantification using microCBA and Qubit assays.” Please cite where the readers can find these results.
Line 314. The authors state: “Vesicles alone and FM 4-64 alone…” What is meant by “EVs alone”?
Line 329. The authors boiled samples to prepare them for NTA. How would this affect the small cellular particles?
The authors outline the cost effectivity in the choice for the method for determination of the amount of EVs in samples. I suggest that they present some estimation of the costs for the methods used (including the necessary equipment).
The authors should discuss the use of recently developed method Interferometric Light Microscopy (ILM) to determine size and number of small particles in samples as an alternative simple and relatively low cost high throughput method:
Boccara, M.; Fedala, Y.; Vénien-Bryan, C.; Bailly-Bechet, M.; Bowler, C.; Boccara, A.C. Full-field interferometry for counting and differentiating aquatic biotic nanoparticles: From laboratory to Tara Oceans. Biomed Opt. Express 2016, 7, 3736–3746.
Romolo, A. et al., Assessment of Small Cellular Particles from Four Different Natural Sources and Liposomes by Interferometric Light Microscopy. Int. J. Mol. Sci. 2022, 23, 15801. https://doi.org/10.3390/ijms232415801
Author Response
Dear Reviewers,
We would like to thank you for your constructive comments and suggestions which have served to improve the content, quality, and clarity of the manuscript. The revised manuscript contains the text revisions and additions as was requested. Specifically, we improved the quality of the introductory material, explanations about the various assays and their costs, as well as added clarifying information to the results and discussion sections. We appreciate the insight and perspective you provided and have addressed each of your specific concerns below.
As there are many types of small cellular particles in the biological samples (e.g. membrane-enclosed vesicles, antibody complexes, lipoproteins, extracellular nucleic acids, cell-engineered particles such as flagella hairs and scales, etc), the authors should state at the beginning what they mean by “extracellular vesicles”. Lipids and proteins may be involved in different types.
By “extracellular vesicles” we are referring to membrane enclosed vesicles. We have clarified that distinction at the beginning of the manuscript.
Did the authors visualize the samples to make sure which types of small cellular particles are present?
The samples in this manuscript were visualized with NTA but not with EM. However, we have worked extensively with vesicles from E. cloacae and have previously published SEM and TEM characterizations of vesicles from this bacterium.
Line 261. NTA measurements did not reproduce well the proportion of EVs related to the dilution of samples. It should be mentioned that EVs do not have fixed identity and can be transformed in number, size and shape if diluted. Please comment how safe is to consider NTA as the reference method.
It has been previously shown that NTA is capable of detecting extracellular vesicles (but not necessarily bacterial EVs) at least when vesicle concentrations are in a specific range. This information as well as comments about the heterogeneity of bEVs has been added to the text (beginning at line 310). The use of NTA as a reference method has also been added to the discussion.
Line 267. The authors state:” For this reason, we tested the correlation between protein concentration and particle quantification using microCBA and Qubit assays.” Please cite where the readers can find these results.
Figures have been added to the end of the sentence (lines 330-332)
Line 314. The authors state: “Vesicles alone and FM 4-64 alone…” What is meant by “EVs alone”?
Text was revised to say “Vesicles without FM 4-64 dye and FM 4-64 dye only samples were also run as controls” (Line 384-388)
Line 329. The authors boiled samples to prepare them for NTA. How would this affect the small cellular particles?
Vesicles were only boiled for protein quantification. The text has been revised to clarify this point (Line 399-405).
The authors outline the cost effectivity in the choice for the method for determination of the amount of EVs in samples. I suggest that they present some estimation of the costs for the methods used (including the necessary equipment).
Nanosight and other assay costs are added at lines 328-336.
The authors should discuss the use of recently developed method Interferometric Light Microscopy (ILM) to determine size and number of small particles in samples as an alternative simple and relatively low cost high throughput method:
Information about ILM and it’s uses have been added to the discussion (beginning at line 321)
Reviewer 3 Report
The authors in the paper “Comparison of methods for quantifying extracellular vesicles of Gram-negative bacteria” reported the comparison of methods for quantifying extracellular vesicles of Gram-negative bacteria. Overall, the topic is interesting but in my opinion the paper cannot be published in this form in “International Journal of Molecular Science”, significant changes must be made to enrich the information contained in this paper and to better understand its focus.
I report my considerations:
· The authors reported that “Vesicles from a wide variety of bacteria are being explored for their roles in bacterial and viral infections and their use as a delivery mechanism for disease treatment or prevention”. In my opinion, a paragraph regarding the different role and possible application of these vesicles should improve the quality of this paper.
· The authors should add functional data, if present, regarding the functional role of the vesicles showing their biological role in order to stress the importance to make a correct quantification.
· In my opinion the authors should report the different methods employed by laboratories to isolate the extracellular vesicles from bacteria. This point could enrich the content of the work and give an overall vision. Moreover, information regarding their characterization should also improve the quality of the paper.
Author Response
Dear Reviewers,
We would like to thank you for your constructive comments and suggestions which have served to improve the content, quality, and clarity of the manuscript. The revised manuscript contains the text revisions and additions as was requested. Specifically, we improved the quality of the introductory material, explanations about the various assays and their costs, as well as added clarifying information to the results and discussion sections. We appreciate the insight and perspective you provided and have addressed each of your specific concerns below.
The authors reported that “Vesicles from a wide variety of bacteria are being explored for their roles in bacterial and viral infections and their use as a delivery mechanism for disease treatment or prevention”. In my opinion, a paragraph regarding the different role and possible application of these vesicles should improve the quality of this paper.
Information about the roles and applications has been added to the discussion (lines 259-271).
The authors should add functional data, if present, regarding the functional role of the vesicles showing their biological role in order to stress the importance to make a correct quantification.
We do not have functional data to add to this manuscript. However, in the added paragraph requested above, we included information regarding the functional roles of bacterial vesicles and discussed the importance of accurate quantification.
In my opinion the authors should report the different methods employed by laboratories to isolate the extracellular vesicles from bacteria. This point could enrich the content of the work and give an overall vision. Moreover, information regarding their characterization should also improve the quality of the paper.
This information has been added to the second paragraph of the introduction.
Round 2
Reviewer 3 Report
The revisions made by the authors improved the quality of the submitted work. In my opinion the paper can be accepted.